# Synthesis and Application of Ion-Exchange Magnetic Microspheres for Deep Removal of Trace Acetic Acid from DMAC Waste Liquid

**DOI:** 10.3390/nano13030509

**Published:** 2023-01-27

**Authors:** Xuna Jin, Yao Lu, Heyao Zhang, Yuheng Ju, Xiaodan Zeng, Xiang Li, Jie Chen, Zhigang Liu, Shihua Yu, Shanshan Wang

**Affiliations:** 1College of Chemical & Pharmaceutical Engineering, Jilin Institute of Chemical Technology, Jilin 132022, China; 2Centre of Analysis and Measurement, Jilin Institute of Chemical Technology, Jilin 132022, China; 3School of Petrochemical Technology, Jilin Institute of Chemical Technology, Jilin 132022, China; 4Steel Making Plant of Jilin Jianlong Iron and Steel Co., Ltd., Jilin 132001, China

**Keywords:** DMAC waste liquid, acetic acid, advanced removal, ion-exchange magnetic microspheres

## Abstract

In order to develop a deep method for removing trace acetic acid from industrial solvents, a type of quaternary ammonium-salt-modified magnetic microspheres was developed as a potential nanoadsorbent for low-concentration acetic-acid-enhanced removal from DMAC aqueous solution. The ion-exchange magnetic microspheres (Fe_3_O_4_@SiO_2_@N(CH_3_)_3_^+^) have been prepared by a two-step sol-gel method with N-trimethoxysilylpropyl-N, N, N-trimethylammonium chloride as functional monomer, tetraethyl orthosilicate as a cross-linking agent, Fe_3_O_4_@SiO_2_ as a matrix. The nanocomposite is characterized by SEM, FI-IR, XRD, VSM, and XPS. Moreover, the optimization of adsorption experiments shows that the maximum adsorption capacity of nanoadsorbent is 7.25 mg/g at a concentration = 30 mg/L, adsorbent dosage = 10 mg, V = 10 mL, and room temperature. Furthermore, the saturated Fe_3_O_4_@SiO_2_@N(CH_3_)_3_^+^ achieved an efficient regeneration using a simple desorption method and demonstrated a good regeneration performance after five adsorption/desorption cycles. In addition, Fe_3_O_4_@SiO_2_@N(CH_3_)_3_^+^ was used to remove acetic acid in DMAC waste liquid; the adsorption effect is consistent with that of a nanoadsorbent of acetic acid in an aqueous solution. These results indicate that Fe_3_O_4_@SiO_2_@N(CH_3_)_3_^+^ can efficiently treat acetic acid that is difficult to remove from DMAC waste liquid.

## 1. Introduction

N, N-dimethylacetamide (DMAC) is an essential solvent for synthetic fibers, and it is usually recycled via extraction and distillation in practical production [1]. DMAC will decompose to a certain extent and produce acetic acid during high-temperature distillation. However, the generated acetic acid and DMAC form azeotrope, and thus a distillation method cannot be used to realize the deep separation of acetic acid and DMAC. Importantly, residual acetic acid will further accelerate the decomposition of DMAC in the next DMAC recovery process. Therefore, the deep removal of acetic acid plays a crucial role in inhibiting the hydrolysis of DMAC. In recent years, few studies have focused on the separation of low-concentration acetic acid from DMAC aqueous solutions. Generally, acetic acid is removed by two main methods. In the first method in Liu’s lab, DMAC waste was diluted with water to 60~80% of the water content, and then sodium hydroxide was added to neutralize the acetic acid in the solution [2]. Next, DMAC was recovered through extraction and distillation from the neutralized aqueous solution. However, it is prone to producing large amounts of salt scales that can corrode easily equipment and cause device blockages [2]. In the second method, Li’s research team adsorbed and removed acetic acid using column chromatography [3]. However, the efficiency of column chromatography technology is low, and thus it has not yet been applied and popularized in industry. In recent years, some studies have been published on the adsorption of acetic acid. Guo and Chen’s research group reported a method of using MOF material UiO-66 to adsorb acetic acid in an aqueous solution [4,5,6]. Yang et al. prepared an alkaline site-modified MgO for the selective adsorption of the CH_3_COOH/C_6_H_6_ binary mixture, and their results show the highest selectivity [7]. Chelazzi et al. synthesized ZnO/castor oil-derived polyurethanes materials, which can be useful in other fields where the efficient capture of acetic acid is necessary [8]. Baglioni et al. prepared zinc oxide-castor oil polyurethane hybrids (ZnO/COPs), which can be used as acetic acid removers in the preventive conservation of cultural heritage [9]. Although they are not used for the removal of acetic acid in a DMAC solvent system, they can be used as a reference for this removal.

Magnetic solid phase extraction technique (MSPE) is a modern separation and purification technology [10,11]. It uses magnetic or magnetic-modified materials as adsorbents and can be directly separated from the matrix via an external magnetic field. Therefore, MSPE is simple to operate, efficient has no centrifugal filtration, and can be widely applied for the extraction and separation of trace pollutants [12,13,14]. The proposed magnetic ion exchange technique is a combination of magnetic solid-phase extraction and ion exchange. It has the dual characteristics of fast exchange speed and fast separation, as well as being inexpensive and convenient to operate. Importantly, it can replace traditional chromatography technology and is widely used to separate contaminants [15,16,17].

In this study, an anion-exchange magnetic microsphere was synthesized using the two-step sol-gel method, its adsorption behavior and regeneration were studied, and a real sample analysis of acetic acid on a nanoadsorbent was conducted. For the first time, a prepared Fe_3_O_4_@SiO_2_@N(CH_3_)_3_^+^ was used to trace acetic acid removal from a DMAC aqueous solution, providing a rapid and low-energy-consumption process for the deep removal of acetic acid from industrial DMAC.

## 2. Materials and Methods

### 2.1. Reagents and Instruments

Fe_3_O_4_ magnetic microspheres (400 nm) were purchased from Jinzhuzhizao Biotechnology Co., Ltd (Jilin, China). Ethanol (AR, ≥99.5%). Acetic acid (AR, ≥99%), tetraethyl orthosilicate (AR, ≥99%), ammonia solution (28.0–30.0%), and methanol (AR, 99.5%) were obtained from Tianjin Damao Chemicals Co., Ltd. (Tianjin, China). N-[3-(Trimethoxysilyl)propyl]-N,N,N-trimethylammonium chloride (50% in methanol) and tetraethyl orthosilicate (TEOS) were purchased from Alfa Aesar Chemical Co., Ltd. (Shanghai, China). Deionized water was obtained from a Millipore Milli-Q apparatus.

Infrared (IR) spectra of the samples were characterized by Fourier-transform infrared (FT-IR) spectrometry (Spectrum One, Perkin-Elmer, Philadelphia, PA, USA). X-ray diffraction (XRD) (D8 FOCUS, Bruker, Salbruken, Germany) with Cu Kα radiation (k = 1.5406 A°) was used to analyze the alteration in the crystallographic phases of magnetic species, and the scanning range was in a 2θ range of 10–80°. Field-emission scanning electron microscopy (FESEM) (Zeiss Marilin, Marilin, Germany) with EDS was used to observe surface morphology and element distribution. The intensity of magnetism was determined by vibrating sample magnetometer (VSM) (703T, Lake Shore, USA). An HPLC analysis was carried out on an LC-20AB system (Shimadzu, Japan). Chromatographic separation analysis was performed on a VP-ODS C18 column (5 μm × 4.6 mm × 150 mm i.d.). The mobile phase consisted of methanol (mobile phase A) and ultrapure water with 0.1% phosphoric acid (mobile phase B), A:B = 5:95. The flow rate was 1.0 mL/min, detection wavelength was monitored at 210 nm, and the injection volume was 20 μL for each input.

### 2.2. Preparation of Fe_3_O_4_@SiO_2_@N(CH_3_)_3_^+^

#### 2.2.1. Preparation of Fe_3_O_4_@SiO_2_ Magnetic Microspheres

First, 0.2 g Fe_3_O_4_, 80 mL anhydrous ethanol, and 20 mL distilled water were added to a round-bottom flask under ultrasonication for 15 min, followed by the addition of 2 mL concentrated ammonia aqueous solution, and the mixture was stirred mechanically for 20 min. Then, 1 mL tetraethoxysilane(TEOS) was added to the above reactor drop-by-drop, and the reaction was stirred for 5 h at room temperature. After complete reaction, the product was washed with water and alcohol, the final product was diluted with water to a concentration of 50 mg/mL [18].

#### 2.2.2. Preparation of Ion-Exchange Magnetic Microspheres

In the next step, 5 mL Fe_3_O_4_@SiO_2_, 150 mL anhydrous ethanol, 25 mL distilled water, 1 mL N-[3-(Trimethoxysilyl)propyl]-N,N,N-trimethylammonium chloride, 2 mL TEOS and 7.5 mL ammonium hydroxide solution were successively added to a 500 mL three-neck round flask, and then the reaction mixture was mechanically stirred at 40 °C for 3 h. Afterward, the ion-exchange magnetic microspheres (Fe_3_O_4_@SiO_2_@N(CH_3_)_3_^+^) were collected by magnetic force, and the product was alternately washed 5 times with water and ethanol to remove the excess reagents, before being dried in a drying oven [19,20].

### 2.3. Adsorption Experiments

The adsorption experiments were performed in triplicate, and the acetic acid solutions used were prepared using standard acetic acid solutions. The adsorption capability and removal efficiency of acetic acid on Fe_3_O_4_@SiO_2_@N(CH_3_)_3_^+^ was calculated using the following equations:(1)$$q_{e}=\frac{\left(C_{0}-C_{e}\right)\times V}{m}$$
(2)$$R_{e}=\frac{C_{0}-C_{e}}{C_{0}}\times 100\%$$
where q_e_ (mg/g) is the adsorption capacity of acetic acid and R_e_ (%) is the removal efficiency, representing the percentage of acetic acid reduced after adsorption. C_0_ and C_e_ (mg/L) are the initial acetic acid initial and equilibrium concentrations, respectively. m (g) is the weight of Fe_3_O_4_@SiO_2_@N(CH_3_)_3_^+^, and V (L) is the volume of phosphate solution.

To optimize the adsorption experimental conditions, batch experiments were conducted in a 25 mL three-mouth flask. The mixtures were shaken at 150 r/min, and the temperature was maintained at 25 ± 1 °C. After reaching equilibrium and then magnetic separation, the residual acetic acid concentration was determined by HPLC-DAD. The effect of the ion-exchange magnetic microspheres dose on the acetic acid adsorption capacity was performed at dosages ranging from 5 mg to 25 mg. Regarding the influence of temperature on adsorption capacity, the mixture was placed in a water bath and shaken at 25 °C, 35 °C, and 45 °C, respectively. Regarding the concentration effect, the experiments employed initial concentrations of 10 mg/L, 20 mg/L, and 30 mg/L, and samples were taken at required time intervals (0–60 min) for acetic acid analysis.

### 2.4. Regeneration Experiments

The exhausted Fe_3_O_4_@SiO_2_@N(CH_3_)_3_^+^ was collected by magnetic separation after a 30 min adsorption period. Afterward, 0.1 M NaOH was mixed and stirred with adsorbents to optimize the desorption solution. Then, the adsorbent was washed with deionized water to neutrality and dried for the next reuse regeneration cycle.

### 2.5. Real Sample Measurement

Fe_3_O_4_@SiO_2_@N(CH_3_)_3_^+^ (2 g and 3 g) was accurately weighed in a 100 mL three-necked flask, added into 50 mL of acetic acid solution with a mass fraction of 0.006–0.06% (60–600 mg/L) and DMAC mass fraction of 10%, and mechanically stirred at room temperature for 20 min. Then, the adsorption capacity was calculated. For comparisons, acetic acid in water was also tested in parallel following the same procedure.

## 3. Results

### 3.1. Characterization of Fe_3_O_4_@SiO_2_@N(CH_3_)_3_^+^

Fe_3_O_4_@SiO_2_@N(CH_3_)_3_^+^ was synthesized and used the nanocomposite to remove acetic acid from DMAC waste liquid via the processes illustrated in Figure 1. Firstly, super-paramagnetic Fe_3_O_4_ was coated with a silica layer using a sol-gel method, resulting in excellent stability and easy surface modification of iron oxide due to the hydrophilic properties of the silica shell. Secondly, TEOS was selected as the coupling agent, and N-[3-(Trimethoxysilyl)propyl]-N,N,N-trimethylammonium chloride as a functional monomer, a layer of quaternary ammonium (QA) was modified on the surface of Fe_3_O_4_@SiO_2_ through a condensation hydrolysis reaction, to form the resulting Fe_3_O_4_@SiO_2_@N(CH_3_)_3_^+^ [21]. Thirdly, the –N(CH_3_)_3_^+^ groups of the nanocomposite were freely exposed to the surrounding solution and interacted with the specific targets, such as acetic acid or other ions in solution, via ion exchange [22] (bottom of the figure).

The morphology of magnetic microspheres is identified by SEM. As shown in Figure 1a, Fe_3_O_4_ nanoparticles exhibit spherical shapes with rough surfaces. After coating with SiO_2_ (Figure 1b), the spherical-shaped Fe_3_O_4_@SiO_2_ has a smooth surface, which makes it easier to modify and enhances its dispersibility in water. Then, a layer of quaternary ammonium (QA) was modified on the surface of Fe_3_O_4_@SiO_2_ to form Fe_3_O_4_@SiO_2_@N(CH_3_)_3_^+^. In Figure 1c, it is clearly shown that quaternary ammonium forms in situ around Fe_3_O_4_@SiO_2_ microspheres with irregular shapes.

Figure 2a shows the FT-IR spectra of Fe_3_O_4_, Fe_3_O_4_@SiO_2_, and Fe_3_O_4_@SiO_2_@N(CH_3_)_3_^+^, respectively. For all the microspheres, a peak at 585 cm^−1^ corresponded to the Fe-O stretching vibration [23]. The peaks located between 1000 and 1250 cm^−1^, as well as those at 798 cm^−1^, assigned to the stretching mode of Si-OH and Si-O-Si in silica [24], respectively, indicating that the silicon layers were successfully coated on the surface of Fe_3_O_4_ by the sol-gel method. In addition, a new peak at 1479 cm^−1^ is assigned to the asymmetric bending modes of the head N-(CH_3_)_3_ methyl group, which is caused by the introduction of quaternary ammonium in the Fe_3_O_4_@SiO_2_@N(CH_3_)_3_^+^ [20,25]. Meanwhile, the declining peak intensity of Fe-O and Si-O-Si in Fe_3_O_4_@SiO_2_@N(CH_3_)_3_^+^ demonstrates the successful aggregations of quaternary ammonium onto the Fe_3_O_4_@SiO_2_. Furthermore, from the XRD results (Figure 2b), the presence of crystal planes with cubic crystal structures proves that all the microspheres are composed of Fe_3_O_4_ [26]. The intensity of the Fe_3_O_4_ characteristic peaks weakens after coating silica layers and quaternary ammonium shells, but the nanocrystalline crystal properties of magnetite particles are not destroyed during the synthesis processes of the Fe_3_O_4_@SiO_2_@N(CH_3_)_3_^+^. The above results are strongly indicative of the successful fabrication of the target product as follows: quaternary ammonium-type ion-exchange microspheres (Fe_3_O_4_@SiO_2_@N(CH_3_)_3_^+^).

The full-survey spectrum of the Fe_3_O_4_@SiO_2_ and Fe_3_O_4_@SiO_2_@N(CH_3_)_3_^+^ are presented in Figure 3, and there are significant peaks at 532.05 eV, 152.19 eV, and 102.54 eV corresponding to O 1 s, Si 2 s, and Si 2 p, respectively [27,28]. Moreover, the high-resolution spectra of Si 2 p and N 1 s are shown in Figure 2b,c. In Si 2 p spectrum, the strength of Si decreases before and after coating quaternary ammonium. Then, the N 1 s spectrum forms a new peak at 402.63 eV, corresponding to the C-N functional groups, due to the introduction of -N(CH_3_)_3_^+^ from quaternary ammonium [29,30]. These results indicate that the synthesis of Fe_3_O_4_@SiO_2_@N(CH_3_)_3_^+^ is successful. Figure 3d shows that the saturation magnetizations of Fe_3_O_4_@SiO_2_ and Fe_3_O_4_@SiO_2_@N(CH_3_)_3_^+^ are 47.85 and 40.49 emu/g, respectively. The composite magnetism decreases due to the non-magnetic layer of -N(CH_3_)_3_^+^ on the surface. Although the magnetic saturation intensity of Fe_3_O_4_@SiO_2_@N(CH_3_)_3_^+^ is reduced, it can easily be separated from the water under an external magnetic field (inset).

### 3.2. Optimization of Absorption Condition by Fe_3_O_4_@SiO_2_@N(CH_3_)_3_^+^

The optimization of the experimental conditions designed by batch adsorption experiments is displayed in Figure 4. As shown in Figure 4a, the increase in the dose of Fe_3_O_4_@SiO_2_@N(CH_3_)_3_^+^ has a negative effect on adsorption because the ratio of acetic acid to the active site decreases. The effect of temperature variation on acetic acid adsorption is shown in Figure 4b. With the temperature increase in the range of 25–45 °C, the adsorption capacity increases from 7.20 to 11.01 mg/g, indicating that acetic acid adsorption is an endothermic process. The influence of acetic acid concentration on the adsorption capacity of synthesized materials (Figure 4c) reveals that, with the increase in concentration from 10 to 50 mg/L, the adsorption capacity increases from 1.81 to 8.67 mg/g. This is due to the presence of vacant sites that are initially more accessible for adsorption; the repulsive forces between the ions dominate and hinder their accessibility. The data show that the adsorption capacity significantly increases in the first ten minutes as the time elapses, with no significant change in the adsorption capacity. Ultimately, under comprehensive consideration, the maximum adsorption capacity is 7.25 mg/g, where the condition of Fe_3_O_4_@SiO_2_@N(CH_3_)_3_^+^ dosage = 1 g/L, initial concentration = 30 mg/L, v = 10 mL, and temperature = 25 °C.

### 3.3. Regeneration Study

For the recovery of Fe_3_O_4_@SiO_2_@N(CH_3_)_3_^+^ after adsorption, 0.1 M NaOH was used as the eluent for elution. After elution, the Fe_3_O_4_@SiO_2_@N(CH_3_)_3_^+^ was collected by a magnet, washed until neutral, dried, and re-adsorbed again. As depicted in Figure 5, after five adsorption–desorption cycles, the adsorption capacity of the nanoadsorbents was only reduced by less than 5%, suggesting that synthetic Fe_3_O_4_@SiO_2_@N(CH_3_)_3_^+^ has an excellent regeneration performance. In addition, the nanomaterials before and after the adsorption of acetic acid remained unchanged in the morphology and crystal structure (Appendix A).

### 3.4. Real Sample Measurement

To have an insight into the practical applicability of synthesized Fe_3_O_4_@SiO_2_@N(CH_3_)_3_^+^, which is applied to extract acetic acid from the DMAC waste liquid. The nanoadsorbent (2 g and 3 g) is added to the aqueous solution and the DMAC waste liquid with an acetic acid concentration of 0.006 and 0.06%, respectively. After full adsorption, the supernatant is assayed by HPLC-DAD, as depicted in Figure 6. When the mass concentration of acetic acid is 0.006% in water and DMAC waste liquid, the removal rate is 100%. Additionally, while the mass concentration of acetic acid increases to 0.06%, the removal rate of acetic acid in water and DMAC waste liquid is the same to be about 60%. This means that the change in the matrix does not affect the removal of acetic acid by Fe_3_O_4_@SiO_2_@N(CH_3_)_3_^+^, and a better removal effect is achieved by increasing the amount of nanoadsorbent (insert picture). All the above conclusions confirm that the Fe_3_O_4_@SiO_2_@N(CH_3_)_3_^+^ is practical for removing acetic acid from DMAC waste liquid.

## 4. Conclusions

In conclusion, a novel quaternary ammonium ion-exchange magnetic microspheres (Fe_3_O_4_@SiO_2_@N(CH_3_)_3_^+^) was synthesized under normal thermal and atmospheric conditions and used for the deep removal of trace acetic acid from industrial solvents. The use of a nanoadsorbent was confirmed by SEM, FTIR, XRD, VSM, and XPS analyses. After optimizing the adsorption conditions through batch experiments, Fe_3_O_4_@SiO_2_@N(CH_3_)_3_^+^ demonstrated the efficient removal of acetic acid. Moreover, ion exchange played a key role in this acetic acid removal. Fe_3_O_4_@SiO_2_@N(CH_3_)_3_^+^ can be effectively regenerated by 0.1 mol/L NaOH and reused five times in adsorption–desorption cycles; the adsorption capacity reduces only by less than 5%. Therefore, the synthesized Fe_3_O_4_@SiO_2_@N(CH_3_)_3_^+^ is potentially applicable for extracting acetic acid from the DMAC waste liquid. Additionally, the extraction result is close to the analyte in an aqueous solution. Therefore, it provides a useful tool for rapidly and effectively removing acetic acid from industrial DMAC via a low-energy process.

## Data Availability

The data presented in this study are available on request from the corresponding author.

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
