# Peer review of "Synthesis and Application of Ion-Exchange Magnetic Microspheres for Deep Removal of Trace Acetic Acid from DMAC Waste Liquid"

_nanomaterials, 2023, doi:10.3390/nano13030509_

Round 1
Reviewer 1 Report
The paper presents the preparation and characterization of a novel nanoabsorbent based on magnetite nanoparticles. The absorption properties of the nanocomposite are achieved by functionalization with an ion-exchange compound. The effective use of the nanocomposite for absorption of low concentration acetic acid in aqueous solution and also in realistic industrial waste water and its magnetic extraction is demonstrated.
Overall, this is very interesting and scientifically sound work that also may become relevant for possible future technological use. From these aspects, I consider this work as suitable for publication in Nanomaterials.
Yet, before this may happen, the presentation needs careful revision:
- - Unfortunately, it is difficult following the text due to problems with English. I urgently recommend to seek help with a natively English speaking colleague, who is also familiar with this kind of research.
- - The structure of the paper should be thought over once more: actually the section Discussion is a mere Conclusion, a proper discussion of the results is missing! One could, e.g., refer to the motivations put forward in the Introduction and discuss the pros and cons of using conventional methods compared with this new method.
- - Check the section headers: those of 2.2 and 3.1 are the same.
- - Check, if all terms are properly defined: e.g. the “removal rate” is not defined (is it a rate? means a quantity per time? I think a language problem?).
In summary, I would support publication, but the authors should consider the above made remarks. After proper revision (when possible misunderstandings from unclear language are removed), I recommend another review concerning scientific details.
Reviewer 2 Report
The paper Synthesis and application of ion-exchange magnetic microspheres for deep removal of trace acetic acid from DMAC wastewater, the authors: Xuna Jin; Yao Lu; Heyao Zhang; Yuheng Ju; Xiaodan Zeng; Xiang Li; Jie Chen; Zhigang Liu*; Shihua Yu*; Shanshan Wang represent an interesting study for Nanomaterials readers, more corrections are necessary.
My principal questions or remarks:
The title is clear.
The content is in accord with title.
The manuscript not adheres to the journal's standards.
The size of the article is appropriate to the contents.
The authors must underline the major findings of their work and explain novelty of this study. The objectives must be better pointed.
The Abstract section refers to the study findings, methodologies, discussion as well as conclusion, but can be completed.
The key words permit found article in the current registers or indexes.
In the introduction isn’t clearly described the state of the art of the investigated problem. 8 articles aren’t enough for demonstrated state of the art of research study. More articles from last years are necessary. It is this study actual?
The methods are well described.
The figures have a good quality.
The comparison with other articles is necessary.
The Conclusion missing from manuscript.
The paper is relatively easy to understand by readers from other area.
The literature is insufficiently critical, current, and internationally evaluated. Please citation references from last years.
The paper was written in standard, grammatically correct English, small corrections are necessary.
Please verify manuscript and respect guide for authors.
The authors must completed the manuscript with:
5. Conclusions
This section is not mandatory but can be added to the manuscript if the discussion is unusually long or complex.
Author Contributions: For research articles with several authors, a short paragraph specifying their individual contributions must be provided. The following statements should be used “Conceptualization, X.X. and Y.Y.; methodology, X.X.; software, X.X.; validation, X.X., Y.Y. and Z.Z.; formal analysis, X.X.; investigation, X.X.; resources, X.X.; data curation, X.X.; writing—original draft preparation, X.X.; writing—review and editing, X.X.; visualization, X.X.; supervision, X.X.; project administration, X.X.; funding acquisition, Y.Y. All authors have read and agreed to the published version of the manuscript.” Please turn to the CRediT taxonomy for the term explanation. Authorship must be limited to those who have contributed substantially to the work reported.
Funding: Please add: “This research received no external funding” or “This research was funded by NAME OF FUNDER, grant number XXX” and “The APC was funded by XXX”. Check careful-ly that the details given are accurate and use the standard spelling of funding agency names at https://search.crossref.org/funding. Any errors may affect your future funding.
Informed Consent Statement: Any research article describing a study involving humans should contain this statement. Please add “Informed consent was obtained from all subjects involved in the study.” OR “Patient consent was waived due to REASON (please provide a detailed justification).” OR “Not applicable.” for studies not involving humans. You might also choose to exclude this statement if the study did not involve humans.
Data Availability Statement:
Reviewer 3 Report
This paper prepared ammonium salt modified magnetic microspheres for deep removal of trace acetic acid from DMAC mixture. This paper assert that this new materials can provide a rapid and low energy consumption process for acetic acid deep removal from industrial DMAC.
However, some parts of contents show questions and logical errors.
- In intro., there is no research trend or efforts. (Ex, OOO group try...) Minimum 3 groups' required.
- In Figure 1., (b) and (c) is not clear. If it has no conductivity, you should coat on surface by carbon or Pt or gold.
- After adsorbent test, you must check samples again. (Ex, SEM, XPS, etc...) This is very important data to understanding adsorption experiments.
Round 2
Reviewer 2 Report
Because the authors improve the manuscript in accord with all recommendations I reconsider initial opinion and accept the paper.
Reviewer 3 Report
This work satisfied accept in this journal